# Renal Cell Carcinoma Unclassified with Medullary Phenotype in a Patient with Neurofibromatosis Type 2

**Sanila Sarkar \*, Whitney Throckmorton, Racheal Bingham, Pavlos Msaouel \*** **, Giannicola Genovese, John Slopis, Priya Rao, Zsila Sadighi † and Cynthia E. Herzog †**

MD Anderson Cancer Care Center, University of Texas, Houston, TX 77030, USA
* Correspondence: ssarkar3@mdanderson.org (S.S.); pmsaouel@mdanderson.org (P.M.)
† These authors contributed equally to this work.

**Abstract:** We present, to our knowledge, the first reported case of germline neurofibromatosis Type 2 (NF2) associated with renal cell carcinoma unclassified with medullary phenotype (RCCU-MP) with somatic loss by immunohistochemistry of the *SMARCB1* tumor suppressor gene located centromeric to NF2 on chromosome 22q. Our patient is a 15-year-old with germline neurofibromatosis Type 2 (NF2) confirmed by pathogenic mutation of c.-854-??46+??deletion. Her NF2 history is positive for a right optic nerve sheath meningioma, CNIII schwannoma requiring radiation therapy and post gross total resection of right frontotemporal anaplastic meningioma followed by radiation. At age 15 she developed new onset weight loss and abdominal pain due to RCCU-MP. Hemoglobin electrophoresis was negative for sickle hemoglobinopathy. Chemotherapy (cisplatin, gemcitabine and paclitaxel) was initiated followed by radical resection. Given the unique renal pathology of a high grade malignancy with loss of SMARCB1 expression via immunohistochemistry, and history of meningioma with MLH1 loss of expression and retained expression of PMS2, MSH2 and MSH6, further germline genetic testing was sent for *SMARCB1* and mismatch repair syndromes. Germline testing was negative for mutation in *SMARCB1*. Therefore, this is the first reported case of RCCU-MP associated with germline *NF2* mutation. This suggests the importance of closer surveillance in the adolescent and young adult population with NF2 with any suspicious findings of malignancy outside of the usual scope of practice with NF2.

**Keywords:** NF2; renal cell carcinoma; medullary phenotype



## 1. Introduction

Neurofibromatosis type 2 (NF2) is an autosomal dominant disorder characterized by increased risk of central nervous system (CNS) tumors, including schwannomas, meningiomas, ependymomas, astrocytomas and neurofibromas [1]. Bilateral vestibular schwannomas are highly suggestive of NF2. Other disease manifestations include peripheral neuropathy, cataracts, retinal hamartomas and cutaneous lesions [2]. The incidence of NF2 is about 1/25,000 with nearly 100% penetrance by age 60 [2]. An abnormal gene is usually inherited from a parent, but the disease may also be manifested by a de novo mutation. The genetic mutation for NF2 is located at chromosome 22q11.2, and encodes a protein called merlin which is a tumor suppressor gene expressed in Schwann cells in the CNS. Management of NF2 involves serial surveillance imaging and a multidisciplinary approach with oncology, neurology, ophthalmology, genetics and neurosurgery. The *SMARCB1* tumor suppressor gene (also known as INI-1, SNF5 or BAF47) is also located at chromosome 22q11.2, centromeric to NF2, and germline *SMARCB1* mutations predispose patients to schwannomas harboring biallelic somatic *NF2* inactivation [3]. Conversely, germline *NF2* mutations are typically not associated with tumors harboring SMARCB1 loss, with the exception of a case report of atypical teratoid rhabdoid tumor (ATRT) harboring somatic SMARCB1 loss in a 3-year-old patient with germline *NF2* mutation [4].

Renal cell carcinoma unclassified with medullary phenotype (RCCU-MP) is a very rare subtype of renal medullary carcinoma (RMC), a highly aggressive renal cell carcinoma characterized by loss of SMARCB1 by immunohistochemistry [5,6]. RMC typically occurs in young individuals of African descent with sickle hemoglobinopathies, such as the sickle cell trait and sickle cell disease [7,8]. In the 2016 WHO classification of renal tumors, expert panel consensus developed the provisional term RCCU-MP for those rare cases of RMC occurring in the absence of a sickle hemoglobinopathy [9]. There are rare case series describing RCCU-MP in adult patients [9], but there are no pediatric patients documented in the literature. We present the first case of germline *NF2* mutation associated with RCCU-MP in an adolescent patient.

## 2. Case Presentation

Our patient is a Hispanic female who first presented at age 2 with clumsiness and dysarthria. Her initial MRI brain/face revealed right vestibular schwannoma but no other NF2-related pathology. Genetic testing was sent from a serum blood sample to the University of Alabama and analyzed for next generation sequencing (NGS) as well as duplication/deletions using multiplex polymerase chain reaction (PCR) assay (MLPA). It revealed a specific pathogenic mutation, c.-854-??46+??deletion, which was previously identified in her father who was diagnosed with NF2. There was no biallelic loss of NF2. Shortly thereafter, she underwent removal of a forehead lesion with pathology confirming a schwannoma. At age 4, a 7-mm enhancing lesion filling the right L1-L2 neural foramen was identified and thought to be a nerve root schwannoma. At age 5, brain MRI revealed a growing right optic nerve/optic nerve sheath that appears slightly fuller in the orbital apex, with concerning meningioma (Figure 1). Since she was not a surgical candidate, she received proton beam radiation therapy to the right optic nerve meningioma and a right cranial nerve schwannoma with a total of 48.6 cGy in 27 fractions without any systemic therapy. At age 8, she was noted to have mild to moderate conductive hearing loss for the right ear, normal hearing sensitivity for the left ear and right middle ear disorder. At age 11, she was noted to have hypothyroidism and started on levothyroxine. She was also being treated for attention deficit hyperactivity disorder (ADHD), anxiety, depression and self-harming behavior.

At age 12, brain MRI revealed that the intracranial meningioma and bilateral vestibular schwannomas were slightly larger. Spine MRI showed multiple thoracic and lumbar spine tumors, most significant at T4 and L2 and L4 cauda equina. There was also a possible ependymoma in the upper cervical spine, but there was no spinal cord compromise from any tumors. Surgery was again not indicated at that time. At age 13, face MRI revealed a large right frontotemporal meningioma that was considerably enlarged from the study three months prior, measuring approximately $5 \times 3 \times 5$ cm compared to approximately $2.2 \times 1 \times 2.4$ cm (Figure 2). There was mass effect of the surrounding brain structures, resulting in effacement of the right lateral ventricle and shift of midline by about 4 mm. She had worsening headaches and vomiting, prompting further imaging that revealed an increase in tumor size. This prompted a gross total resection via craniotomy of the right temporal meningioma. Pathology of the tumor confirmed an anaplastic meningioma of WHO grade III. Genetic testing on tumor demonstrated MLH1 loss of expression with focal positivity and retained expression of PMS2, MSH2 and MSH6. Due to the high-grade meningioma, she proceeded with radiation and completed 3600 Gy. Towards the end of the year, she returned to surgery to resect a nerve sheath tumor on the right side of the neck, with pathology confirming schwannoma.

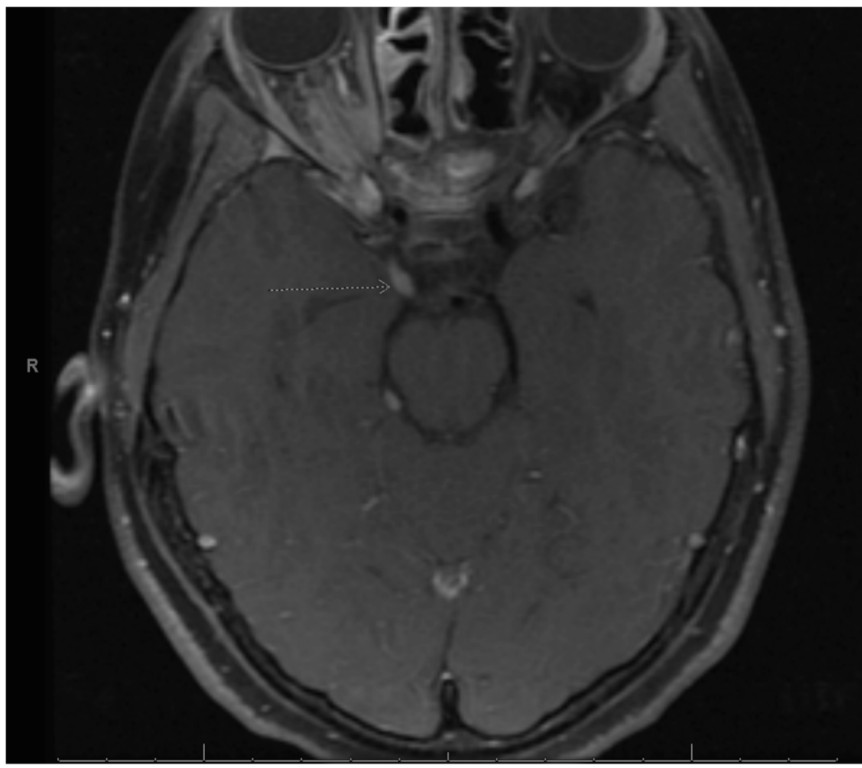

**Figure 1.** Right optic nerve sheath tumor. Arrow is pointing to the right optic nerve sheath tumor as stated in the figure title.

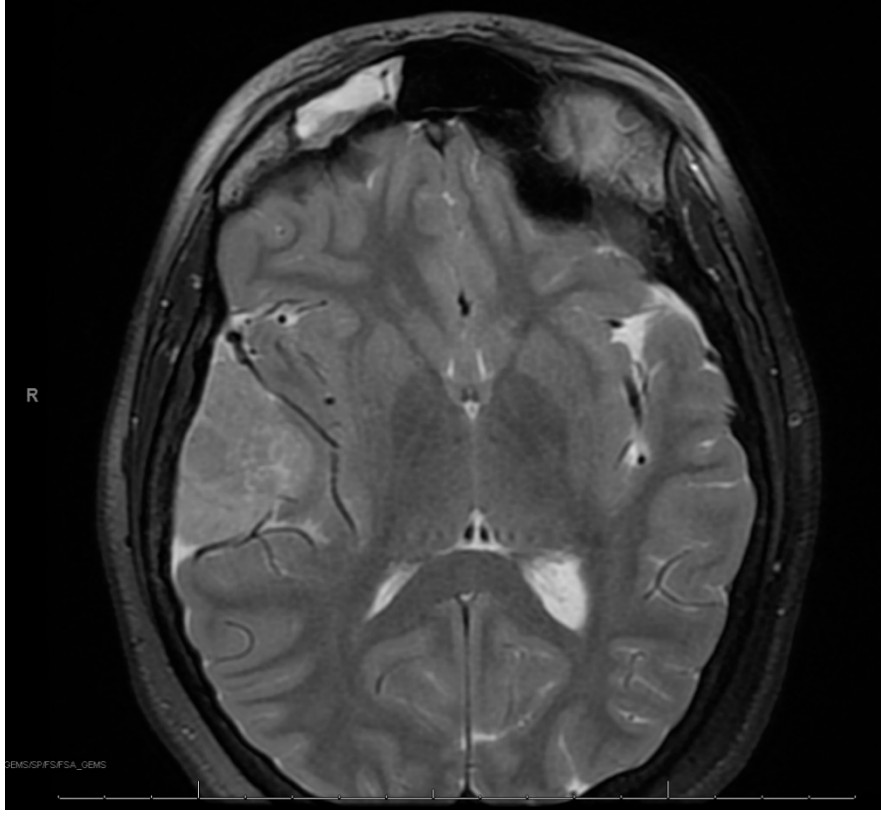

**Figure 2.** Right frontotemporal anaplastic meningioma.

At age 15, she presented to an outside ER with a 2–3-day history of worsening right lower quadrant abdominal pain, nausea and occasional vomiting. She also had a history of a 20-lb weight loss over the past few months due to decreased appetite. CT scan showed a right renal mass 7 cm in size. She was transferred to our hospital and MRI abdomen/pelvis showed an indeterminate left adrenal nodule (Figure 3) and a large, solid, peripherally enhancing right renal mass with central heterogeneity measuring 8.1 × 5.4 × 8.7 cm (Figure 4). It also showed retroperitoneal lymphadenopathy with a retrocaval node at 4.1 × 2 cm, superior retrocaval node at 2.6 × 1.6 cm and an inferior precaval/aortocaval node at 1.5 × 0.9 cm. There was also a right lung nodule noted on a chest CT (Figure 5). She then had a baseline positive emission tomography (PET) scan which again showed the right renal mass, concerning for primary renal cell carcinoma, FDG avid retroperitoneal lymphadenopathy, concerning for metastasis and small bilateral pulmonary nodules, concerning for metastasis. She underwent a biopsy of the right renal mass, which was inconclusive. She underwent a second biopsy and was found to have RCCU-MP manifesting as a high-grade malignancy with loss of SMARCB1 by immunohistochemistry in the absence of a sickle hemoglobinopathy.

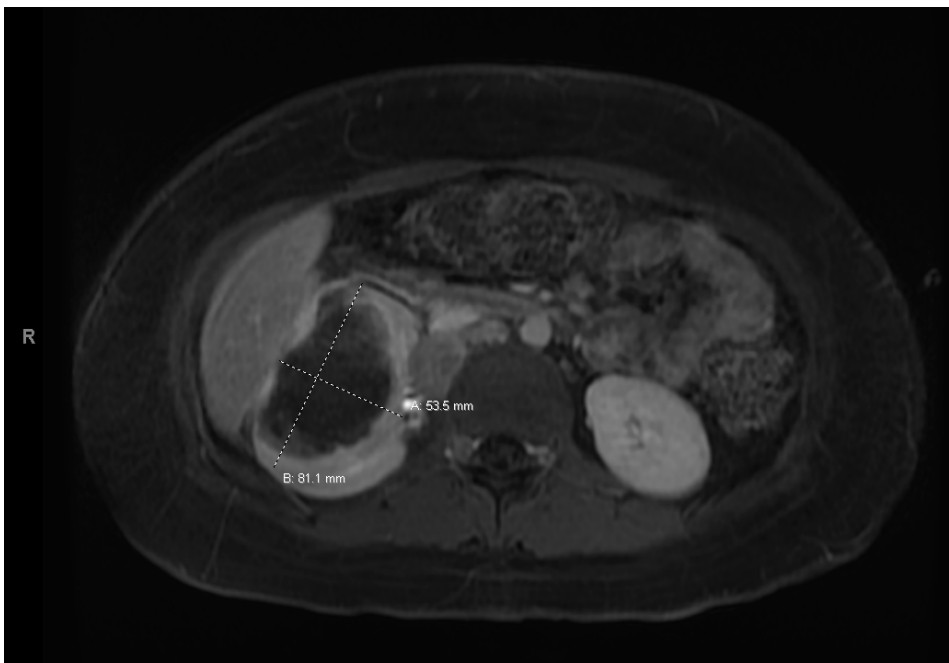

**Figure 3.** Left adrenal nodule.

Given the unique renal pathology associated with SMARCB1 loss by immunohistochemistry staining, and the history of meningioma with MLH1 loss of expression with focal positivity and retained expression of PMS2, MSH2 and MSH6, further germline genetic testing for mismatch repair syndromes was performed that included additional germline pathogenic variants *BAP1*, *CDC73*, *CDKN1C*, *DICER1*, *DIS3L2*, *EPCAM*, *FH*, *FLCN*, *GPC3*, *MET*, *MLH1*, *MSH2*, *MSH6*, *PMS2*, *PTEN*, *REST*, *SDHB*, *SDHC*, *SMARCA4*, *SMARCB1*, *TP53*, *TSC1*, *TSC2*, *VHL* and *WT1*, all of which were negative.

After a multidisciplinary discussion, this patient was started on a chemotherapy regimen containing cisplatin 70 mg/m$^2$ on day 1, gemcitabine 1000 mg/m$^2$ on days 1, 8 and 15 and paclitaxel 80 mg/m$^2$ on days 1, 8 and 15, which is a regimen often used in patients with RMC [10,11]. As per the latest recommendation on RMC management [5], the plan was for neoadjuvant chemotherapy followed by delayed surgery for total resection.

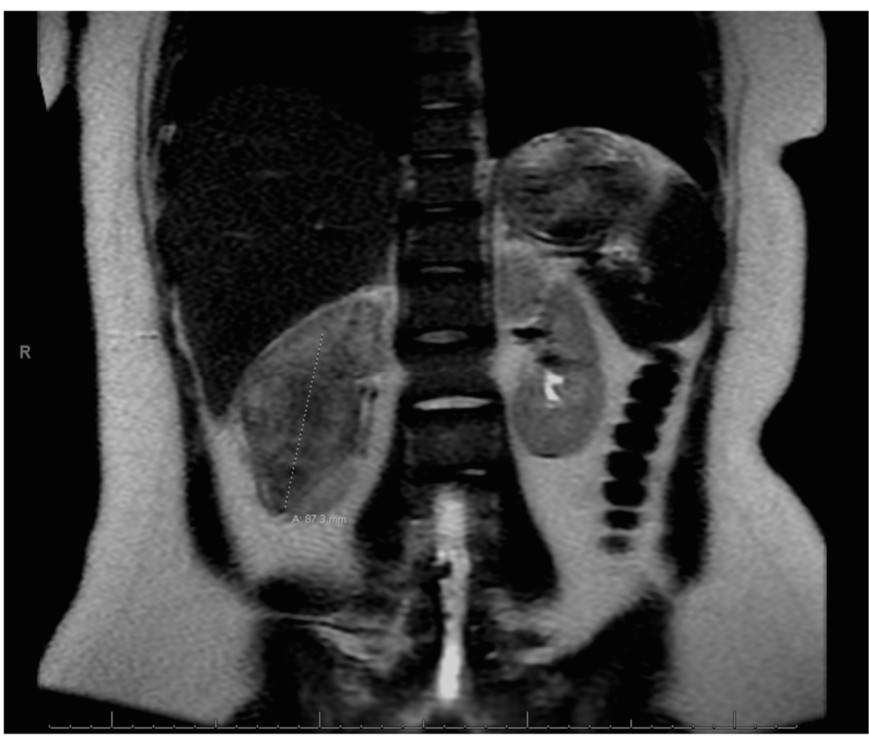

**Figure 4.** Right renal tumor.

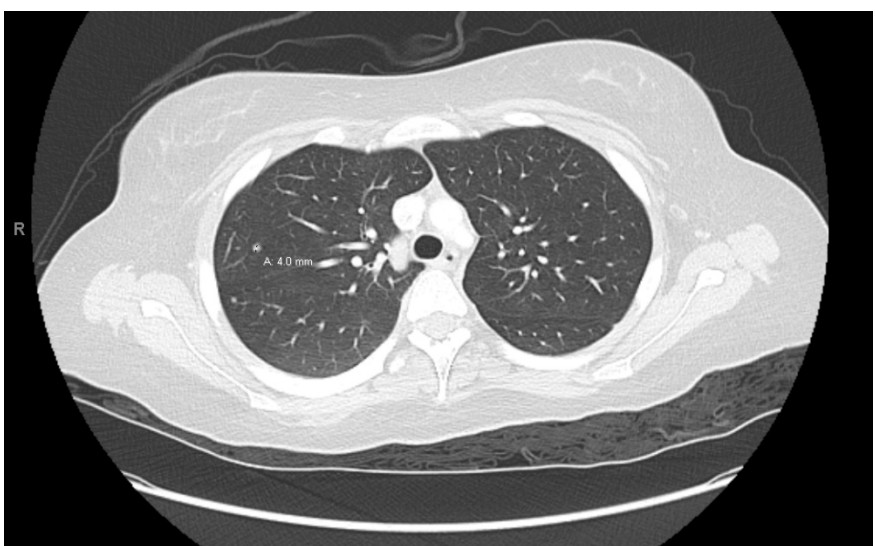

**Figure 5.** Right lung nodule.

She began treatment with the above regimen and experienced multiple complications during treatment. Most significantly, she developed *Clostridium difficile* after her first cycle, and remained an inpatient for a few weeks due to significant dehydration, hyponatremia, hypokalemia and hypomagnesemia. She continues to remain on oral replacements for sodium, potassium and magnesium. Thrombocytopenia was noted on day 15 of chemotherapy, and the decision was thus made to omit day 15 treatment in both cycles 1 and 2 and to reduce gemcitabine to 75% of planned dose after cycle 1.

A follow-up CT scan after 2 cycles showed that the right renal mass had decreased to $6.9 \times 6.6$ cm. Treatment response was noted in the lymph nodes, with a retrocaval node that was $4 \times 2.1$ cm having decreased to $1.9 \times 2.2$ cm, while a $1.7 \times 1.1$ cm node was stable. There was interval decrease in size of the bilateral pulmonary nodules/metastases. Also seen on this CT was an indeterminate calcification in the thoracic spinal canal ($0.9 \times 0.6$ cm) and several

scattered indeterminate punctate sclerotic lesions in pelvic bones, bilateral proximal femurs, the spine, sternum, scapula and proximal humerus.

She completed 2 more cycles of chemotherapy and then proceeded with a right nephrectomy and retroperitoneal lymph node sampling. Pathology was consistent with excellent treatment response and no evidence of viable tumor in either the nephrectomy or the seven resected paracaval and aortocaval lymph nodes. After surgical clearance, she resumed chemotherapy. However, carboplatin 400 mg/m$^2$ on day 1 was substituted for cisplatin due to reduction of kidney function. Cycle 5 day 15 of chemotherapy was delayed for thrombocytopenia, and day 15 was omitted from cycle 6 again due to thrombocytopenia. Towards the end of treatment, she required prolonged antibiotic therapy for multiple infections. She was diagnosed with enteropathogenic *E. coli* gastroenteritis and *E. coli* urinary tract infection and was treated with IV cefepime. She had increased abdominal tenderness and was diagnosed with typhlitis based on CT abdomen. Antibiotics were escalated to IV meropenem. During the same hospitalization, blood cultures were positive for *Candida tropicalis*, and *Staphylococcus epidermis*. IV vancomycin and IV caspofungin were added to her treatment. She completed a course of oral linezolid and oral fluconazole to complete a course of 14 days. Her blood counts recovered, and she was able to complete a total of 6 cycles of chemotherapy, with end of therapy PET/CT negative for any FDG-avidity and no evidence of residual tumor, as well as abdomen MRI negative for disease (Figure 6).

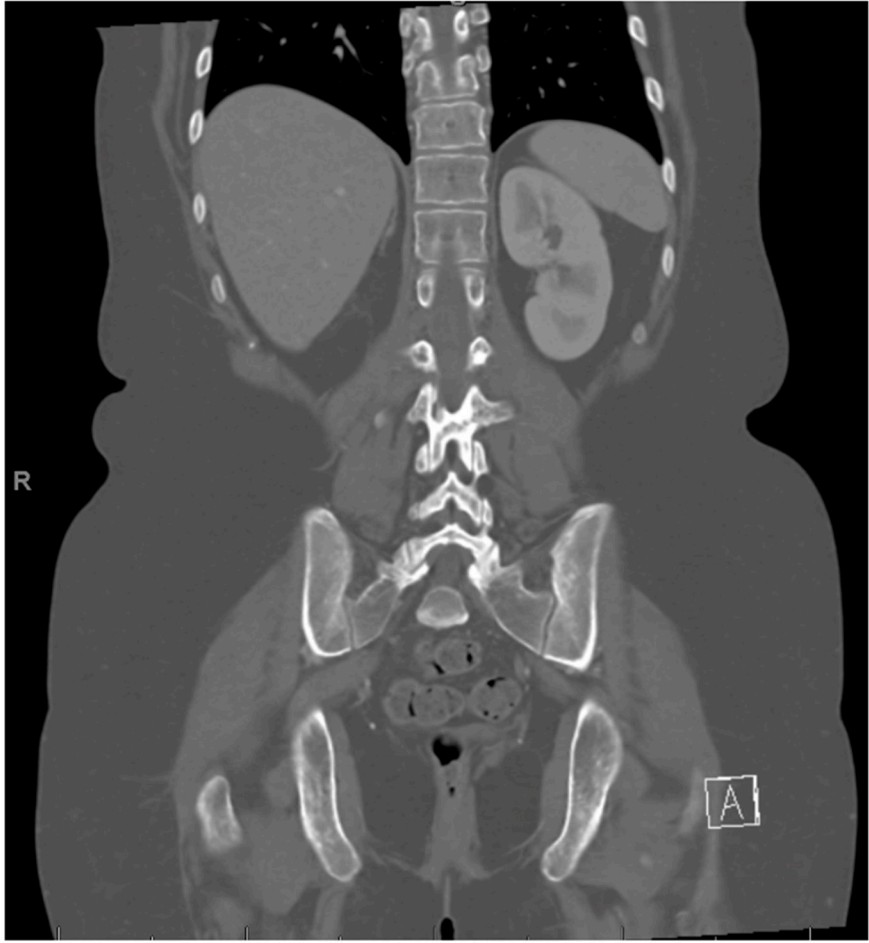

**Figure 6.** Post right nephrectomy.

Our patient continues to do well from an oncology standpoint, with no evidence of recurrent renal tumor. She also has imaging confirming stable neuro-oncology disease. She continues to follow with nephrology for chronic kidney disease (CKD) stage 2/3a,

and hypertension in the setting of a solitary kidney. She has bilateral mild to severe high frequency sensineural loss that has been stable and is followed regularly by audiology. She has hormone deficiencies secondary to prior radiation therapy and is followed regularly with endocrinology. Nutrition is optimized with our dietician. She continues to work with physical therapy and occupational therapy. She has completed neuropsychological testing and has resumed school studies. Our team continues to offer a multidisciplinary approach to medical care, including psychosocial support.

## 3. Pathology

The biopsy of right kidney demonstrated mostly fibrinous material with small fragments of renal parenchyma with interstitial chronic inflammation and no definite tumor. The biopsy of a retroperitoneal lymph node showed high grade tumor composed of discohesive rhabdoid cells with highly atypical nuclei and eosinophilic cytoplasm involving lymphoid tissue with areas of necrosis and neutrophilic inflammation with necrosis and limited viable tumor cells (Figure 7). By immunohistochemistry, the cells were positive for pankeratin (Figure 8) and PAX8 (Figure 9) and negative for CK7, CK20, GATA3, CDX2, TTF1, CAIX, ERG, S100, CD34, CD30, SMA, desmin, OCT3/4 and CD163. The tumor cells showed diffuse loss of SMARCB1/INI-1 (Figure 10). The INI-1 was reviewed by several experienced pathologists that interpreted the stain as negative. While there is some background cytoplasmic and nuclear staining in the entire specimen, the norm is to compare staining with the adjacent internal positive control. Given that, in this case, the lymphocytes stained strongly positive for INI-1, it was appropriate to interpret this is as a negative stain in this context. As evidenced by the high-power image, the staining appears to be of a background variety and limited to the cytoplasm, in which the nuclei remain negative. Hence, this is interpreted as a negative stain.

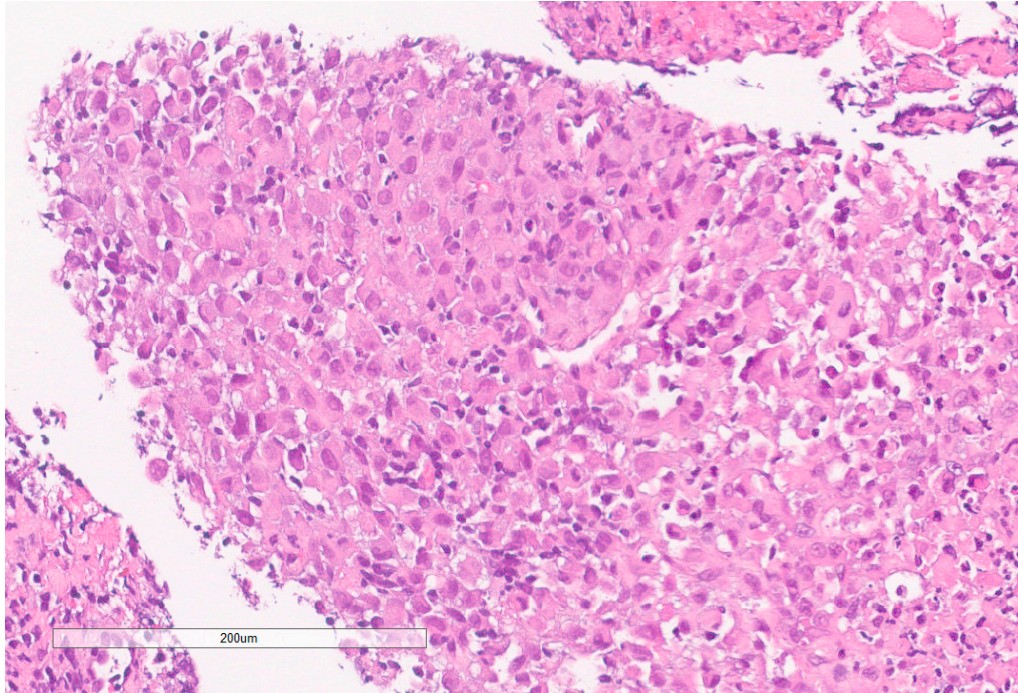

**Figure 7.** High grade carcinoma with necrosis; tumor cells have a rhabdoid morphology.

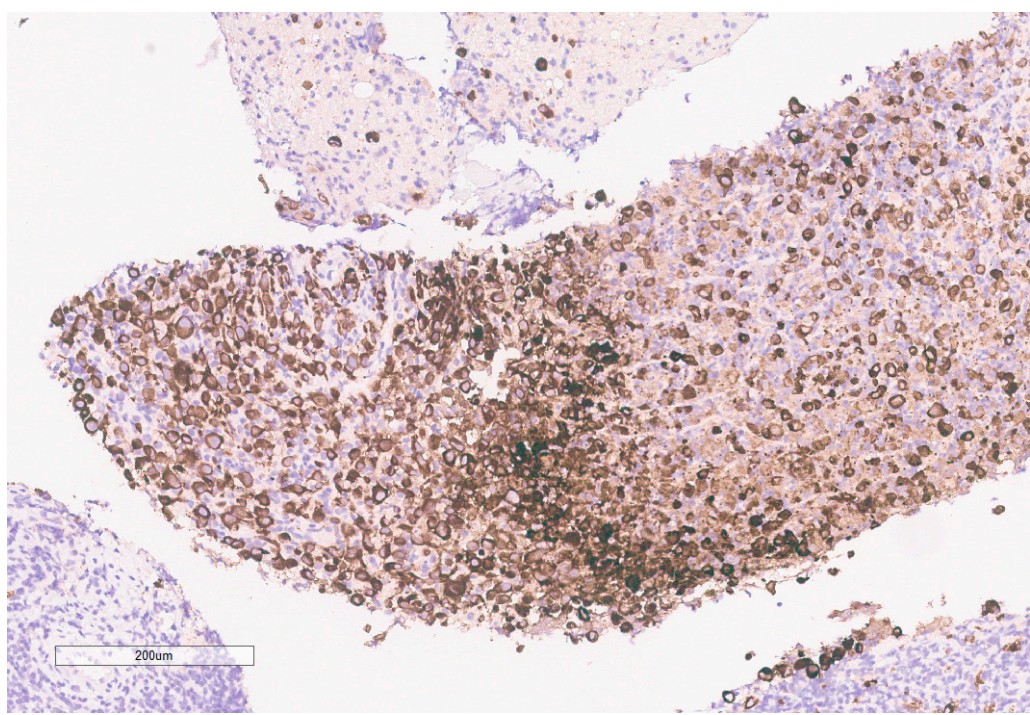

**Figure 8.** Pancytokeratin stain is positive in the tumor.

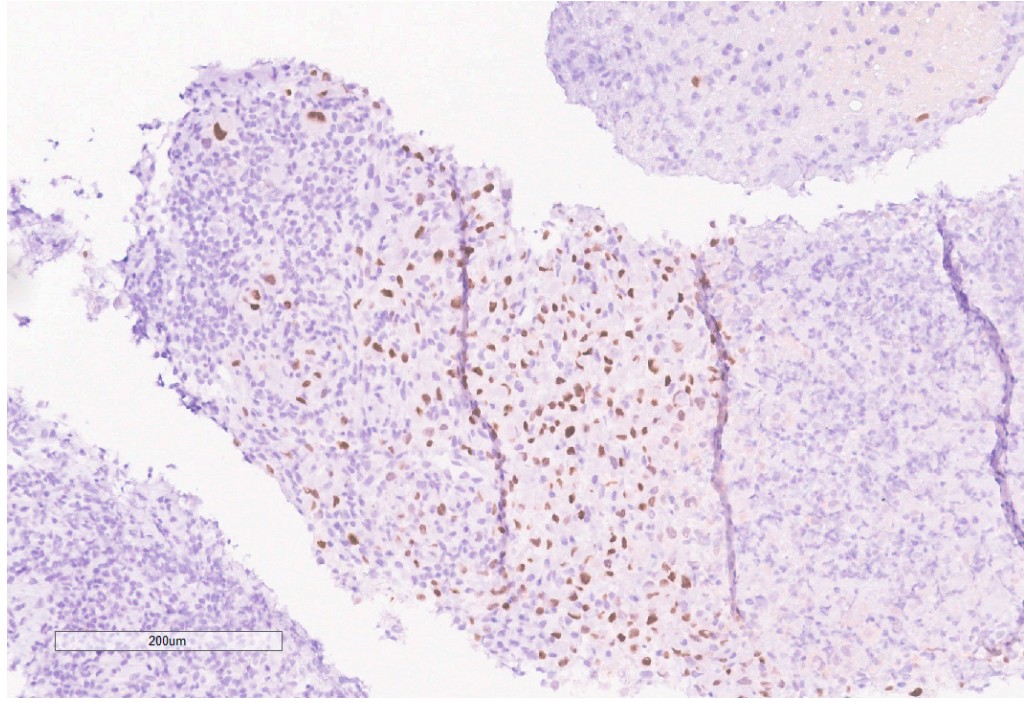

**Figure 9.** PAX8 is positive in the tumor.

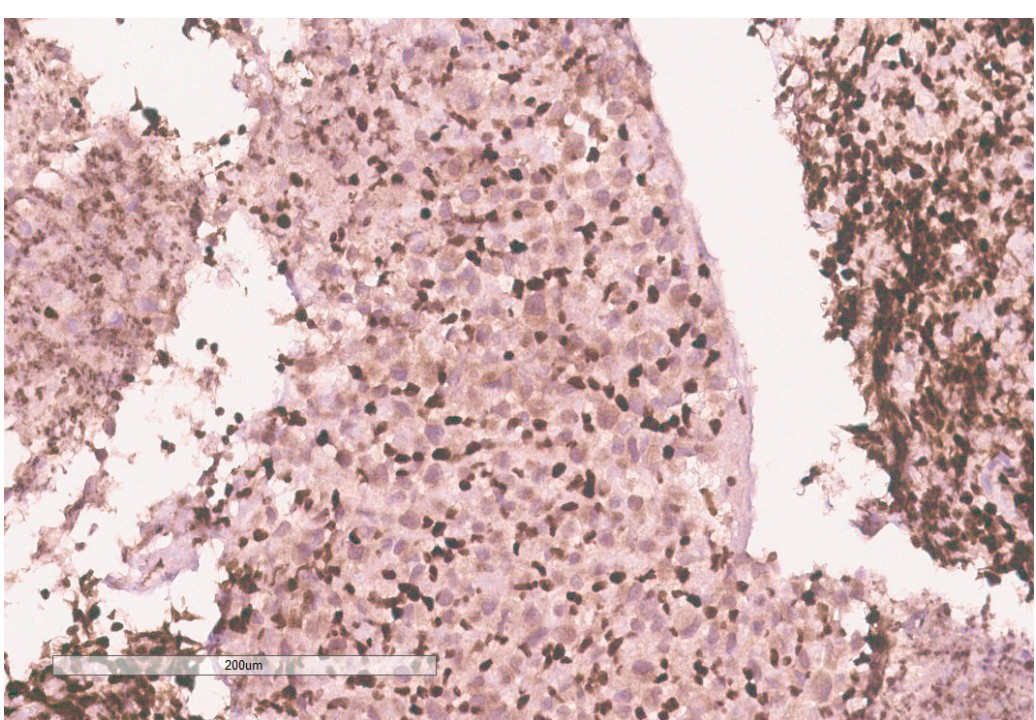

**Figure 10.** INI-1 (BAF47) is negative in the tumor. Lymphocytes serve as an internal positive control.

Given the expression of PAX8, and loss of SMARCB1 in the neoplastic cells, a diagnosis of RMC was favored. Malignant rhabdoid tumor (MRT) of the kidney was also considered due to the rhabdoid morphology in the setting of SMARCB1 loss. However, MRTs are typically negative for PAX8 and our specimen showed diffuse PAX8 staining. Furthermore, MRTs typically occur in children younger than 3 years old [11]. Hemoglobin electrophoresis was negative for sickle cell traits or other detectable hemoglobinopathies favoring the diagnosis of the RCCU-MP subtype of RMC. NGS of the tumor specimen did not identify copy number variants or somatic mutations. This NGS based its analysis for the detection of somatic mutations on the coding sequence of 134 genes and selected copy number variations or amplifications in 47 genes, for a total analysis of 146 genes with overlap. This was performed on DNA extracted from the sample in a clinical laboratory improvement amendment (CLIA) certified molecular diagnostic laboratory. This genetic analysis included analysis of *NF2*.

## 4. Discussion

Our patient represents a very unique oncologic case. Renal tumors are not usually screened for in NF2 patients. Although rare case reports show RCC with somatic mutations of *NF2* in adult populations [12,13], RCC is very unusual in this age group and typically not associated with germline NF2 disease. There have previously been no case reports of RMC or its RCCU-MP variant in a patient with germline NF2 disease. Sestini et al. postulated a 4-hit mechanism involving the tumor suppressor genes *SMARCB1* and *NF2*. They identified a de novo germline deletion/insertion in the *SMARCB1* gene in a patient with schwannomatosis [14]. Three different tumors derived from this patient showed deletion/insertion and a somatic *NF2* mutation on the same allele, but no other *SMARCB1* mutations. In addition, two of the tumors had somatic loss of heterozygosity encompassing the *SMARCB1* and *NF2* region in chromosome 22q11.2. In tumor tissues from two other patients, they found a somatic *SMARCB1* or *NF2* mutation in association with loss of heterozygosity, but no germline mutations were identified. They proposed that this 4-hit mechanism was the etiology for development of tumors. Kotch et al. recently attributed this 4-hit mechanism in the pathogenesis of SMARCB1-deficient ATRT in the setting of germline *NF2* mutation [4]. A similar process may have resulted in our case of SMARCB1-deficient

RCCU-MP in a young patient with germline *NF2* mutation. The lack of somatic alterations detected by our NGS assay is consistent with prior experience in RMC and RCCU-MP, whereby clinical grade NGS is unable to identify the genomic causes of *SMARCB1* loss in the majority of cases [4,15].

RMC and its RCCU-MP variant are typically highly aggressive malignancies with a median survival of only 13 months from diagnosis and a 29% response rate to platinum-based chemotherapy [5,16]. Our patient demonstrated a profoundly rare complete response to platinum-based chemotherapy. It is unclear whether this gratifying response is related to the unique genetic background of this particular case. Of note, the ATRT case reported by Kotch et al. had a very aggressive response and was refractory to intensive cytotoxic chemotherapy [4].

In conclusion, we present the first case of pediatric RCCU-MP in a patient with *NF2* germline mutation. She has responded well to treatment similar to RMC treatment and is currently disease free. This tumor is not typical for patients with NF2 and is outside the scope of routine surveillance. However, the close proximity of *NF2* and *SMARCB1* within chromosome 22q11.2 and their established synergy in the development of other neoplasms [3] suggests that SMARCB1-deficient malignancies such as RMC/RCCU-MP, ATRT and potentially MRT, should be high in the differential diagnosis in patients with germline *NF2* mutations presenting with tumors outside the usual spectrum associated with NF2. In these scenarios, pathological evaluation should include staining for SMARCB1 loss.

**Author Contributions:** Conceptualization by S.S., W.T., R.B., P.M., G.G., J.S., P.R., Z.S. and C.E.H.; writing—original draft preparation, S.S., W.T. and R.B.; writing—review and editing by S.S., W.T., R.B., P.M., G.G., J.S., P.R., Z.S. and C.E.H. All authors have read and agreed to the published version of the manuscript.

**Funding:** This study was supported in part by the Cancer Center Support Grant to MDACC (grant P30 CA016672) from the National Cancer Institute. Pavlos Msaouel is supported by a Career Development Award by the American Society of Clinical Oncology, a Research Award by KCCure, the MD Anderson Khalifa Scholar Award, the Andrew Sabin Family Foundation Fellowship, a Translational Research Partnership Award (KC200096P1) by the United States Department of Defense, an Advanced Discovery Award by the Kidney Cancer Association, a Translational Research Award by the V Foundation, the MD Anderson Physician-Scientist Award, and philanthropic donations by the family of Mike and Mary Allen. Pediatric Solid Tumors Comprehensive Data Resource Core (Grant # RP180819) and the MD Anderson Cancer Center Institutional Tissue Bank.

**Informed Consent Statement:** This project has received waiver of informed consent and authorization by the University of Texas MD Anderson Cancer Center Institutional Review Board (protocol # 2023-0081).

**Data Availability Statement:** Data sharing not applicable. No new data were created or analyzed in this study. Data sharing is not applicable to this article.

**Conflicts of Interest:** Pavlos Msaouel has received honoraria for service on a Scientific Advisory Board for Mirati Therapeutics, Bristol Myers Squibb, and Exelixis; consulting for Axiom Healthcare Strategies; non-branded educational programs supported by Exelixis and Pfizer; and research funding for clinical trials from Takeda, Bristol Myers Squibb, Mirati Therapeutics, Gateway for Cancer Research, and UT MD Anderson Cancer Center. Cynthia Herzog served on a Data Monitoring Committee for Merck Sharp & Dohme Corp.

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
