# Peer review of "Renal Cell Carcinoma Unclassified with Medullary Phenotype in a Patient with Neurofibromatosis Type 2"

_curroncol, doi:10.3390/curroncol30030255_

Round 1

Reviewer 1 Report

Please provide more detail on the NF2 germline variant c.-854-??46+??deletion (i.e., genomic location, reference genome used). What is the supportive evidence used to determine pathogenicity for this particular 5’ noncoding region variant?

Please define what the acronym AYA represents.

Please provide reference for “NF2 is about 1/25,000 to 1/40,000” (pg 1, line 30).

Italicize all gene names.

Fig 9 looks as though there is faint nuclear staining in the tumor nuclei. The tumor is also immediately adjacent to an area of necrosis where lymphocytes show diminished, but intact nuclear expression. From anecdotal experience, BAF47 staining can be heterogeneous and somewhat diminished but considered intact. Is this how the BAF47 typically performs for your immunohistology lab when loss of expression is seen? I have doubts that this represents true loss of expression.

Please describe the NGS panel used for the tumor specimen including number and names of genes included, platform used, and capability of detecting heterozygous gene loss and translocation.

It is stated that NGS of the tumor did not identify copy number variants or somatic mutations. Did this panel include NF2? Was there bi-allelic loss of NF2 in the tumor?

Author Response

Hello and thank you for your edits! Please see attachment!

Reviewer 2 Report

Excellently written with a complete description of the case provided by the authors. The only additional recommendation I suggest is if the authors provide additional photomicrographs of the CNS tumors to help the audience appreciate the overlapping diseases described in this case (ie NF2 and RCCU-MP). This is not required but is strongly recommended. 

Author Response

Hello and thank you for your edits! I am able to add additional radiographic images of CNS tumors, but unable to add photomicrographs from pathology of tumors.